

# Bedload transport measurements with Swiss impact plate geophones in two Austrian mountain streams (Fischbach and Ruetz): system calibration, grain size estimation, and environmental signal pick-up

Dieter Rickenmann[1], Bruno Fritschi[1]

[1]Swiss Federal Research Institute WSL, Birmensdorf, 8903, Switzerland

*Correspondence to*: Dieter Rickenmann (dieter.rickenmann@wsl.ch)

**Abstract.**

The Swiss plate geophone system is a bedload surrogate measuring technique that has been installed in more than 20 streams, primarily in the European Alps. Here we report about calibration measurements performed in two mountain streams in Austria. The Fischbach and Ruetz gravel–bed streams are characterized by important runoff and bedload transport during the snowmelt season. A total of 31 (Fischbach) and 21 (Ruetz) direct bedload samples were obtained during a six year period. Using the number of geophone impulses and total transported bedload mass for each measurement to derive a calibration function, results in a strong linear relation for the Fischbach, whereas there is only a poor linear calibration relation for the Ruetz measurements. Instead, using geophone impulse rates and bedload transport rates indicates that two power law relations best represent the Fischbach data, depending on transport intensity; for lower transport intensities, the same power law relation is also in reasonable agreement with the Ruetz data. These results are compared with data and findings from other field sites and flume studies. We further show that the observed coarsening of the grain size distribution with increasing bedload flux can be qualitatively reproduced from the geophone signal, when using the impulse counts along with amplitude information. Finally, we discuss implausible geophone impulse counts that were recorded during periods with smaller discharges without any bedload transport, and that are likely caused by vehicle movement very near to the measuring sites.

## 1 Introduction

In the past decade or so, an increasing number of studies were undertaken on bedload surrogate acoustic measuring techniques which were tested both in flume experiments and in field settings. A review of such indirect bedload transport measuring techniques was recently published by Rickenmann (2017a, 2017b). Examples of measuring systems include the Japanese pipe microphone (Mizuyama et al., 2010a, 2010b; Uchida et al. 2013; Goto et al. 2014), the Swiss plate geophone (Rickenmann and Fritschi, 2010, Rickenmann et al. 2012, 2014), other impact plate systems (Krein et al., 2008, 2016; Møen et al. 2010; Reid et al. 2007; Beylich and Laute 2014; Taskiris et al. 2014), and hydrophones, i.e. underwater microphones (Barton et al. 2010; Camenen et al. 2012; Rigby et al. 2015). It is well known that bedload transport rates often show very



large variability for given flow conditions (Gomez, 1991; Leopold and Emmett, 1997; Ryan and Dixon, 2008; Recking,
2010), and that prediction of (mean) bedload transport rates is still very challenging, particularly for steep and coarse–
bedded streams (Bathurst et al., 1987; Nitsche et al., 2011; Schneider et al. 2015, 2016). For such conditions, direct bedload
transport measurements are typically difficult to obtain, or may be impossible to make during high flow conditions (Gray et
al., 2010). In contrast, indirect bedload transport measuring methods have the advantage of providing continuous monitoring
data both in time and over a cross–sections, even during difficult flow conditions, and are therefore expected to increase our
understanding of bedload transport.
A fair number of these measuring techniques have been successfully calibrated for total bedload flux, which generally
requires contemporaneous direct bedload transport measurements in the field (Thorne, 1985, 1986; Voulgaris et al., 1995;
Rickenmann and McArdell, 2007, 2008; Mizuyama et al. 2010b; Rickenmann et al. 2014; Mao et al., 2016; Habersack et al.
2016; Kreisler et al. 2016). Essentially, linear or power law relations were established between a simple metric
characterizing the acoustic signal and bedload mass. In some studies further calibration relations were established to identify
particle size, either based on signal amplitude (Mao et al., 2016; Wyss et al., 2016a) and/or on characteristic frequency of
that part of the signal which is associated with a single impact of a particle (e.g. for impact plate systems; Wyss et al., 2016b)
or by determining a characteristic frequency for an entire grain size mixture (for the hydrophone system; Barrière et al.,
2015a). A few of the acoustic measuring techniques were used to determine bedload transport by grain size classes (Mao et
al., 2016; Wyss et al., 2016a). Finally, some studies examined to what extent findings from flume experiments can be
quantitatively transferred and applied to field sites for which independent, direct calibration measurements exist (Mao et al.,
2016; Wyss et al., 2016b, 2016c).
In this study we report on calibration measurements of the Swiss plate geophone (SPG) system in two mountain streams
in Austria. The Fischbach and Ruetz gravel–bed streams are characterized by important runoff and bedload transport during
the snowmelt season. During a six year period, 31 (Fischbach) and 21 (Ruetz) direct bedload samples were obtained in the
two streams, respectively. The objectives of this paper are: (i) to present and discuss different ways of analysing the
geophone calibration measurements, also in comparison with data and findings from other field sites and flume studies; (ii)
to show that the observed coarsening of the grain size distribution with increasing bedload flux can be qualitatively
reproduced from the geophone signal; and (iii) to discuss implausible geophone impulse counts that were recorded during
periods with small discharge and without any bedload transport, and that are probably associated with close–by vehicle
movement.

## 2 Field sites and calibration measurements

### 2.1 Overview of field sites and geophone measurements

The first indirect bedload transport measurements using impact plates were made in the Erlenbach from 1986 to 1999 using a
piezoelectric crystal as sensor, with the aim of continuously monitoring the intensity of bedload transport and its relation to



stream discharge (Bänziger and Burch, 1990; Rickenmann, 1994, 1997; Hegg et al., 2006; Rickenmann and McArdell,
2007). A geophone sensor was used at the Erlenbach and at all other field sites that were set up in the year 2000 and later
(Rickenmann and Fritschi, 2010). In the meantime, the SPG system has been installed in more than 20 streams primarily in
Central Europe (Rickenmann, 2017b). An array of steel plates is typically installed flush with the surface of a sill or check
dam, a location where there is only a small chance for (substantial) deposition of bedload grains during transport conditions.
The Fischbach and Ruetz field sites were installed by the Tyrolean Hydropower Company (TIWAG). They are located in
partly glaciated catchments in the Tyrolean Alps (Fig. 1), at elevations of 1544 m a.s.l. (Fischbach) and 1688 m a.s.l.
(Ruetz). Thus, the streams have a nival runoff regime, with typical daily discharge variations and regular bedload transport
during snow and glacier melt in spring and summer. At both field sites, water discharge and bedload transport have been
monitored since 2008. The stream cross–section is essentially trapezoidal at both measuring sites, with the banks protected
by riprap and inclined at 45° (Fig. 2, 3). The geophone sensors are fixed in a cylindrical aluminium case and are mounted on
the underside and in the middle of stainless steel plates, which are screwed into supporting steel constructions (UPN profiles)
and are acoustically isolated by elastomer elements. The steel plates are 0.360 m long, 0.496 m wide, and 0.015 m thick. The
entire steel construction is 8.2 m long (transvers to the flow direction) and embedded into a concrete sill, founded two meters
into the river bed. The entire concrete structure is 8.7 m wide, and it is laterally inclined at 5% to the river left side (Fig. 3),
which improves the discharge measurements at low flows. The sill is protected with riprap on the up- and downstream side.
Starting with the first steel plate located 0.35 m from the right bank, every second steel plate is equipped with a geophone
sensor, so that there are a total of eight sensors at each site.
At both sites, the concrete sill is located 4 m downstream of the cross–section where flow stage is measured on the left
side of the stream, and where flow velocity measurements are made by TIWAG to establish a flow rating curve. At the
Fischbach, a bridge crosses the stream some 13 m upstream of the concrete sill, and provides vehicle access to the measuring
hut on the left side on a small forest road with very infrequent traffic. Along the right side of the stream a local paved road
passes nearby, situated only in 5 m horizontal and about 4.5 m vertical distance above the concrete sill with the geophone
plates (Fig. 3). Uphill the road leads to the village of Gries with about 200 inhabitants. This is the only village to be accessed
upstream of the measuring site. In winter it serves as a relatively small ski resort. At the Ruetz, a bridge crosses the stream
some 15 m upstream of the concrete sill, and provides vehicle access to a large parking lot, paved with gravel, on both sides
of the stream. The measuring site is located at Mutterbergalm in the Stubai valley. From there a cable car provides access to
a large skiing area (winter) and to a hiking area (summer) in the mountains. The public road ends at Mutterbergalm. The
parking lot is situated in a similar minimal distance to the measuring cross section as at the Fischbach, i.e. with about 5 m
horizontal and 4.5 m to 5 m vertical distance above the concrete sill. This information is important for the analysis and
interpretation of the pick-up of geophone signal by environmental sources other than bedload transport.





## 2.2 Direct bedload measurements for system calibration

At each of the two sites, a streamlined metal pillar was installed 0.5 m downstream of the plate with the geophone sensor no. 5 to facilitate the calibration measurements. The metal pillar has a height of 2.5 m and a maximum width of 0.25 m and ensures that a pressure–difference type metal basket sampler fits snugly onto the bed and can be held in place during the bedload sampling operation (Fig. 2b). The aperture of the basket is 50 cm by 50 cm, the same width as the sensor plate. The basket has a notch (cut–out) at a downstream distance of 0.45 m from the aperture (Fig. S1, Supporting information). The notch is somewhat larger than the cross–section of the metal pillar, and the inside of the notch is equipped with rollers. This system allows an exact positioning of the basket during geophone calibration measurements. The maximum width of the basket is 0.90 m and the total length is 2.10 m. During operation the upper surface of the sampler is horizontal while the lower surface is declined at 15% in the downstream direction, in line with the artificial bed in the vicinity of the metal pillar. Over the 0.80 m tail–end of the sampler, the top and sidewall surfaces of the basket are made of 10 mm metal wire mesh. The total volume of the basket is about 0.91 m$^3$.

The calibration measurements used here were obtained by TIWAG in both streams during the summer months of 2008 – 2013 using the basket bedload sampler. A total of 31 measurements from the Fischbach and 21 measurements from the Ruetz were used in this analysis (Table 1). The maximum sample mass caught in the sampler was 518 kg (including particles finer than 10 mm) in the Fischbach; assuming a bulk density of 1600 kg m$^{-3}$, the bedload volume of this sample was about 0.32 m$^3$ or about a third of the total sampler volume. Four calibration measurements from the Fischbach could not be used due to overfilling of the sampler. The grain size distribution of the samples was determined by sieve analysis by a TIWAG–owned engineering consultant. A line–by–number analysis was performed in both streams in October 2012 to estimate the grain size distribution of the bed surface upstream of the geophone sites.

## 2.3 Signal pre–processing, recorded geophone values, and amplitude histogram analysis

The bedload impact shocks on the steel plate are transmitted to the geophone sensor and, thereby, an electrical potential is produced. The standard geophone sensor uses a magnet in a coil as an inductive element. The magnet picks up the vibrations of the steel plate and induces a current in the coil which is proportional to the velocity of the magnet. Whenever the voltage exceeds a preselected threshold amplitude value, $A_{min}$, the shock is recorded as an impulse. Contrary to all the other sites equipped with an SPG system, the threshold amplitude value $A_{min}$ used to determine *IMP* values was set at 0.07 V at the Fischbach and Ruetz (Tables 2, 3). The reason is that the first regular geophone recordings in the Fischbach had shown maximum amplitudes in excess of 10 V, the upper limit of the recording system. To increase the resolution of large amplitudes, the raw signal was dampened by about 30%. To compensate for lower signal strength in relation to the impulse counts, the threshold amplitude value $A_{min}$ was also reduced by 30% when compared with a typical value of 0.1 V used at other sites.



At most of the field sites with SPG measurements, several signal summary values were routinely stored in the past. The
most often used summary value for calibration purposes are the summed impulse counts *IMP*. These values were found to
correlate reasonably well with bedload mass or volume transported (Rickenmann and McArdell, 2007, 2008; Rickenmann et
al., 2012, 2014). Another useful summary value is maximum amplitude *MaxA* that may be determined for different recording
intervals. During calibration measurements, all summary values were typically stored in 1 second intervals. During normal
flow monitoring, the recording interval for the summary values at the Fischbach and Ruetz was 15 minutes. (At other SPG
measurement sites operated by WSL this recording interval is typically 1 minute).
Using the so–called amplitude histograms (AH), Wyss et al. (2016, 2014) demonstrated for the SPG measurements at the
Erlenbach (Swiss Prealps) that absolute bedload masses for each grain size class could be successfully calculated for both the
calibration and validation data obtained with the moving basket samplers. The continuous recording of AH data was also
implemented at the Fischbach and Ruetz measuring sites, with a recording interval of 1 minute. At these sites, impulses were
determined separately for 17 amplitude classes as listed in Table 2. For the analysis in this study, for each amplitude
threshold value $A_{\text{th}}$ (upper class boundary value) a corresponding particle size *D* was estimated according to an empirical
relation given in Wyss et al. (2016c, Eq. 11) and reported in Appendix A as Eq. (A1).
**3. Results**
**3.1 Calibration relations for bedload mass and bedload flux using impulse counts**
The following calibration relations and calibration coefficients were determined using the transported bedload mass *M*, for
particles with *D* larger than 10 mm, the impulses *IMP* summed over the sampling period of duration $T_s$:
$M = k_{\text{lin}}\, IMP$ (1)
$M = k_{\text{pow}}\, IMP^e$ (2)
$k_{\text{tot}} = \Sigma M / \Sigma IMP$ (3)
where the units are in [kg] for *M* and for the coefficients ($k_{\text{lin}}$, $k_{\text{pow}}$, $k_{\text{tot}}$), and the $\Sigma$ sign implies a summation over all the
calibration measurements per site. Equations (1) and (2) were obtained from a linear regression (using log values in case of
Eq. 2), while Eq. (3) represents a mean, linear calibration coefficient based on the total mass and the total number of
impulses for all calibration measurements taken together. The resulting coefficients ($k_{\text{lin}}$, $k_{\text{pow}}$, $k_{\text{tot}}$), exponents (e) and
statistical properties of the calibration relations are reported in Table 3. The squared correlation coefficient $r^2$ was
determined between the measured masses *M* and the estimated masses $M_{\text{reg}}$ (using eq. 1, 2, or $k_{\text{tot}}$ in eq. 2). The relative
standard deviation $s_{\text{e,r}}$ is determined for the ratios ($M_{\text{reg}}/M$), using the regression relation to determine $M_{\text{reg}}$ from the recorded
impulses *IMP*.



For the Fischbach, the calibration relations in the form of Eqs. (1) and (2) show a rather high correlation coefficient (Fig.
4, Table 3), which is also characteristic for similar calibration relations determined for the Erlenbach (Rickenmann et al.,
2012, 2014). For the Ruetz, the calibration relations in the form of Eqs. (1) and (2) are less well defined (Fig. 5, Table 3).
Due to the inclusion of four additional calibration measurements obtained in 2012 and 2013, the correlation coefficient for
the Ruetz is lower than in an earlier analysis that used only 17 measurements from the period 2008 to 2011 (Rickenmann et
al., 2014). This level of correlation is similar to calibration measurements obtained for the Navisence stream in Switzerland
(Wyss et al., 2016c) for which most measured bedload masses were smaller than 20 kg; for the Ruetz, 15 out of 21
calibration measurements also have bedload masses smaller than 20 kg. Using the $k_{tot}$ coefficient from Eq. (3) in Eq. (1)
results in very similar statistical properties as compared to using $k_{lin}$ in Eq. (1), also with a slightly poorer performance
(Table 3).
Systematic flume experiments were performed for different grain size classes to investigate the dependence of a linear
calibration coefficient, defined as $k_{bj} = IMP/M$, on grain size $D$ (Wyss et al., 2016b). This study used bedload particles from
four streams including the Ruetz and Fischbach, and it was found that $k_{bj}$ values showed a local maximum at a grain size $D$
of around 40 mm, in agreement with earlier flume experiments using quartz spheres of different diameters (Rickenmann et
al., 2014). Therefore, we analysed the field calibration measurements from the Ruetz and Fischbach in a similar way (Fig. 6),
and these data essentially confirmed the findings from the flume experiments. The bedload samples from the Ruetz and
Fischbach show a general tendency for $D_{84}$ to increase with increasing unit bedload transport rate $q_b$ (Fig. 7), where $D_{84}$ is
the grain size for which 84 % of material by weight are finer (determined for particles with $D > 10$ mm). It is therefore not
surprising that $k_{bj}$ values also exhibit a local maximum when plotted against the impulse rate, $IMPT$ (Fig. 8), which is a
proxy for transport rate, and where $IMPT = IMP/(T_s \, w_p)$, with the plate width $w_p = 0.5$ m. Finally this lead us to determine
alternative calibrations in terms of unit bedload transport rate per plate width $q_{b,p}$ as a function of impulse rate, $IMPT$ (Fig.
9), with a limiting value of around 0.5 to 1 ($0.5^{-1}$ m$^{-1}$ s$^{-1}$) to separate the two ranges with a different power law function:
$q_{b,p} = a_1 \, IMPT^{b1}$        for $IMPT < 0.48$ ($0.5^{-1}$ m$^{-1}$ s$^{-1}$)                                    (4)
$q_{b,p} = a_2 \, IMPT^{b2}$        for $IMPT > 0.48$ ($0.5^{-1}$ m$^{-1}$ s$^{-1}$)                                    (5)
where the units for $q_{b,p}$ are in (kg $0.5^{-1}$ m$^{-1}$ s$^{-1}$) and for $IMPT$ in ($0.5^{-1}$ m$^{-1}$ s$^{-1}$), and the coefficients and exponents are given in
Table 3. Here, we determined $q_{b,p}$ and $IMPT$ deliberately per unit width of one plate since using the traditional 1 m unit
width would result in different coefficients $a_1$ and $a_2$ (and a different threshold value $IMPT$ separating the application range
of Eq. 4 and Eq. 5), which would entail the risk of erroneous transformations of measured $IMPT$ values into $q_{b,p}$ values for
each plate.
In Fig. 9, the regression relation for higher impulse rates was derived based on 14 calibration measurements from the
Fischbach with $IMPT > 1$ [(1/0.5) m$^{-1}$ s$^{-1}$]. Similarly, the regression relation for lower impulse rates was derived based on 17
measurements from the Fischbach and 19 measurements from the Ruetz, all with $IMPT < 1$ [(1/0.5) m$^{-1}$ s$^{-1}$]. The two power
law relations intersect at $IMPT = 0.48$ [(1/0.5) m$^{-1}$ s$^{-1}$]. Using this limiting value, they were applied to the Fischbach and



 Ruetz data, resulting in the statistical properties of the calibration relations (4) and (5) as reported in Table 3. It appears that

the data from both channel sites can be described reasonably well with these calibrations relations, the relative standard
deviation $s_{e,r}$ being about 98% for the higher impulse rates and about 110% for the higher impulse rates (Table 3). If Eqs.
(4) and (5) are applied to all calibration measurements of each stream separately, the clearly better statistical properties result
for the Fischbach ($r^2 = 0.97$, $s_{e,r} = 61$ %) than for the Ruetz ($r^2 = 0.50$, $s_{e,r} = 145$ %). In comparison to the calibration relation
determined with Eq. (2) for the Fischbach, Eqs. (4) and (5) will predict larger bedload transport rates for very small or very
large *IMPT* values (Fig. 9).

### 193 3.2 Coarsening of grain sizes with increasing bedload flux reflected in geophone signal

The amplitude histograms (AH data) for each calibration measurement were used to estimate grain size distributions (GSD)
for the basket sampler measurements, which were then compared with the sieve analyses of the bedload samples. For the
analysis of the AH data, the lowest class with impulses for $A_{max} < 0.056$ V was excluded, as this class represents
predominantly signal noise. For the remaining 16 classes the sum of the impulses per amplitude class was determined for all
1 min time steps for the duration $T_s$. This resulted in the proportion of impulses per amplitude class per calibration
measurement, not yet weighted for grain size. The impulses per class were weighted by the geometric mean diameter of each
class (Table 2) to the $2^{nd}$ power, $D_m^2$, to estimate the cumulative distribution of AH-values; this weighting procedure
corresponds essentially to the method of Wyss et al. (2016), which is summarized in Appendix A. It is also noted that the
start (and end) time of the bedload sampling does not exactly correspond to the start (and end) time of the recorded AH data,
which introduced a further (generally minor) uncertainty when interpolating AH data for the first and last recording time step
of each bedload sampling period. For the results shown in Figures 10 and 11, the GSD was averaged for given classes of unit
bedload transport rates $q_b$, assigning the same weight to each measurement in a given $q_b$ class. Bedload transport classes and
corresponding abbreviation names are defined in Fig. 10 and 11.
For the bedload samples from both Fischbach and Ruetz a general coarsening trend of the grain size distribution (GSD)
with increasing unit bedload transport rate $q_b$ can be observed, in agreement with general bedload transport theory (Parker,
2008). However, GSDs from individual calibration measurements are quite variable within given classes of $q_b$, both for the
bedload samples and for the estimated GSD from the AH values, and do not necessarily follow the general trend. The GSDs
estimated from the AH values generally show a qualitatively similar trend as the GSDs from the direct bedload samples, but
with a limited quantitative agreement between the two methods.
For the Fischbach (Fig. 10) it is noted that only 2 calibration samples were available for the class Fi1, and these had the 2
smallest bedload masses (with 19 and 8 kg, respectively); this may be a reason for the poor agreement between estimated
and measured GSDs. Similarly, the largest $q_b$ class Fi4 for the Fischbach includes only 1 bedload sample. For the Ruetz (Fig.
11) we note that for the classes Ru1 and Ru3 the bedload masses were relatively small, including only 5 to 6 kg. Together
with a small number of bedload samples (3 and 2, respectively), this may again be one reason for the relatively poor





agreement between estimated and measured GSDs. In contrast, the bedload masses for the Ruetz for the class Ru2 (11 to 23
kg) and Ru4 (15 to 129 kg) were clearly larger.

**3.3 Environmental noise pick-up of the geophone signal**

Both measuring stations are situated at a relatively high elevation, and the stream catchments include mountain peaks with
elevations above 3000 m a.s.l. Therefore the runoff during the winter period is very low, with a base flow below 0.6 $m^3 s^{-1}$ at
the Fischbach and below 0.3 $m^3 s^{-1}$ at the Ruetz. During such flow conditions, only about half or two thirds of the sill with
the steel plates is submerged under water (Fig. 2, Fig. S2). However, during winter geophone impulses are regularly
recorded at all the geophone sensors in both streams (Fig. 12, Fig. 13). According to hydraulic calculations and observations
the sill becomes fully submerged for flows of about 2.5 $m^3 s^{-1}$ at the Fischbach and about 2.0 $m^3 s^{-1}$ at the Ruetz. Therefore it
is unlikely that these geophone impulses are the result of bedload transport.
For the Fischbach and the discharge classes smaller than 3 $m^3 s^{-1}$ the mean *IMP* values per 15 minutes ($IMP_{15}$) vary
between about 0.3 and 2.0. A similar analysis as in Fig. 12 but with a finer discharge resolution (classes of 0.25 $m^3 s^{-1}$) is
presented in Fig. S3. It is also obvious that plates (sensors) no. 1 to 3 generally recorded more impulses than the other plates
no. 4 to 8 (Fig. 12, Fig. S3), which is unlikely a result of bedload transport. For discharges up to about 3 $m^3 s^{-1}$ traffic noise
appears to be a likely source of the geophone impulses, since the local road passes on the river right side very close to the
plates no. 1 to 3 (Fig. 2). For discharge classes larger than 4 $m^3 s^{-1}$ the plates no. 4 to 8 (which have a larger water depth than
plates no. 1 to 3) start to record more impulses on average ($IMP_{15}$) than plates no. 1 to 3; in addition the $IMP_{15}$ values start to
increase with increasing discharge (Fig. 12, Fig. S3). This behaviour is more in line with expectations from bedload–
transport induced signals.
For the Ruetz and the discharge classes smaller than 1.0 $m^3 s^{-1}$ the mean $IMP_{15}$ values vary between about 0.2 and 2.0.
Plates no. 5 to 8 generally recorded more impulses than the other plates no. 1 to 4 (Fig. 13, Fig. S4). The plates no. 1 to 3 are
typically not submerged during these flow conditions, and no signal is to be expected from bedload transport. Again, traffic
noise appears to be a likely source of the measured geophone impulses. The plates on the river left side (5 to 8) tend to
register more impulses on average because the access road to the parking lot passes on this side, hence more parking traffic
is to be expected. A clearer dominance of the plates no. 5 to 8 (which have a larger water depth than plates no. 1 to 4)
becomes apparent for discharge classes larger than about 1.5 $m^3 s^{-1}$ at the Ruetz (Fig. 13, Fig. S4), which is in line with
expectations from bedload–transport induced signals. The mean value of $IMP_{15}$ averaged over all eight plates becomes larger
than about 2 for discharges larger than roughly 2.0 $m^3 s^{-1}$, and above this discharge level the $IMP_{15}$ values start to increase in
general with increasing discharge.
To further investigate the potential source of the implausible geophone recordings, we classified the measured $IMP_{15}$
values into 15 minute intervals during each day–time (Figs. S5, S6). For both streams and low flows, there is a clear daily
cycle of geophone impulse activity although discharge remains rather constant during the entire day. This pattern clearly is
present for the Fischbach for discharges $Q$ smaller than about 3 $m^3 s^{-1}$ and for the Ruetz for $Q$ smaller than about 1.5 $m^3 s^{-1}$.





Geophone activity is higher during the afternoon and the first half of the night at the Fischbach, and primarily during day
time at the Ruetz. A clear absence of this or a similar daily pattern is evident for the Fischbach for $Q$ larger than about 6 m$^3$ s$^{-1}$
and for the Ruetz for $Q$ larger than about 3.5 m$^3$ s$^{-1}$ (Fig. S5, S6). This is a further indication that the geophone impulses at
smaller discharges are mainly traffic induced. Taken together, the above analysis and interpretation suggests that bedload
transport may be the dominant source of producing geophone impulses above a critical discharge $Q_c$ of about 3.5 m$^3$ s$^{-1}$ at the
Fischbach, and above a $Q_c$ of about 1.5 m$^3$ s$^{-1}$ at the Ruetz.
Turowski et al. (2011) analysed the start and end of bedload transport in gravel-bed streams, including geophone
measurements from the Fischbach and Ruetz for the years 2008 and 2009. Based on the above delineation of the $Q_c$ values
for the two streams, it is estimated that they used about 62 % (out of 95 measurements) potentially implausible values for the
Fischbach and about 41 % (out of 492 measurements) potentially implausible values for the Ruetz. If these values were
discarded from their analysis, this would change the histograms of the discharge at the start and end of transport for the two
streams but it would not affect the general conclusions of the study.
**4. Discussion**
**4.1 Calibration relations for the Swiss plate geophone system and grain size determination**
For a system such as the Swiss plate geophone it is known that the signal response depends on factors such as grain size,
fluid or particle velocity, particle shape and mode of transport (i.e. sliding, rolling, saltating), and impact angle and impact
location on the steel plate (e.g. Wyss et al., 2016b; Rickenmann, 2017b). For a given stream we may assume that the most of
these factors vary within a given range, and the linear calibration coefficients primarily vary with flow conditions. Therefore,
we expect that the mean signal response from a given particle size traveling over the plate becomes more stable the larger is
the total number of particles that have been transported over the plate. This is the main reason why we have primarily
considered the summed geophone summary values in the past (e.g. Rickenmann et al., 2012, 2014). Calibration
measurements from various sites confirmed the expectation that random factors influencing the signal response tend to be
more averaged out for longer integration periods (Rickenmann and McArdell, 2007, 2008; Rickenmann et al., 2012, 2014;
Wyss et al., 2016a, 2016c).
However, it may also be interesting to consider calibration relations for example between bedload rates and impulse
rates. If a linear calibration relation in the form of Eq. (1) is generally valid, a division of $M$ and $IMP$ by the sampling
duration $T_s$ to determine rates will typically result in similar values for the linear calibration coefficient. Having performed
this alternative analysis in terms of bedload rates and impulse rates for the data of this study, two distinctly different ranges
of geophone signal response were found based on the data from the Fischbach (Fig. 9). These calibration measurements
suggest that two power law calibration relations in terms of rates provide a better fit than a single linear calibration relation
for the entire domain. The existence of two different ranges is likely a result of a changing GSD with increasing bedload
transport rates. We therefore also plotted data from calibration measurements at many other sites (Fig. 14), but no clear trend





for a similar pattern can be observed for most of these sites. The only exception is the Urslau stream in Austria; the
individual calibration measurements for this stream indicate a trend for a power law relation between $q_b$ and *IMPT* with an
exponent b < 1 for smaller $q_b$ values and with an exponent b > 1 for lager $q_b$ values (Kreisler et al., 2016). These calibration
measurements cover a range of about three orders of magnitude of $q_b$ values; however different methods were used to obtain
the bedload samples for smaller and larger bedload transport intensities, and for the smaller range of $q_b$ values the number of
measurements is limited.

289        We used the AH data recorded during the calibration measurements at the Fischbach and Ruetz to estimate the

transported bedload mass for each calibration measurement, $M_{est}$, by applying the procedure presented by Wyss et al.
(2016a). This method is summarized in Appendix A, and it was specifically developed for the measuring conditions at the
Erlenbach stream in Switzerland. Here, we used Eq. (A3) with the coefficient and exponent determined from the Erlenbach
measurements; the relation of Eq. (A3) is expected to vary somewhat from site to site, and its application here is therefore
associated with uncertainty. To assess the performance of this procedure when applied to the Fischbach and Ruetz, we
plotted the ratio of estimated to observed bedload mass, $M_{est}/M$, as a function of bedload transport rate per plate $q_{b,p}$ and of
observed mass $M$ (Fig. 15). There is generally an over–estimation of bedload mass, up to a factor of about 10. Interestingly,
the over–estimation decreases with increasing bedload transport rate (Fig. 15a). This result is in agreement with Fig. 14,
which suggests that site–specific differences for calibration relations in terms of bedload transport rates and impulse rates
tend to be relatively smaller for higher values of $q_b$. The degree of over–estimation of bedload mass as well as the scatter
around a mean trend line for both streams appears to decrease also with increasing bedload mass for the data of the
Fischbach and Ruetz (Fig. 15b), but this trend is somewhat less pronounced. Concerning grain size estimation from bedload
surrogate measuring techniques, it may be noted that only a few other acoustic measuring techniques were (partly) successful
in determining bedload transport by grain size classes from field measurements (Barrière et al., 2015b, using an impact plate
hydrophone system; Mao et al., 2016, using a Japanese impact pipe microphone system).

305        To illustrate the uncertainty associated with using different calibration relations, we determined the yearly bedload (*YBL*)

for 2010, which represents the year with the largest peak discharges and the largest *YBL* values (Table 4) for the period
2008–2013. For both streams, the *YBL* values are larger when using Eqs. (4) and (5) as compared to using Eq. (1); this is not
surprising when comparing the linear with the power law calibration relations in Fig. 9. The power law calibration relations
result in a 66 % higher *YBL* for the Fischbach and in a 85 % higher *YBL* for the Ruetz, if only plausible *IMP* values for
discharges larger than $Q_c$ are considered; the differences are larger if the entire *IMP* data set for 2010 is considered,
including many implausible values recorded during low flow periods (Table 4). The between–stream comparison shows a
much larger *YBL* for the Fischbach than for the Ruetz, which is due to more frequent peak discharges in the Fischbach
exceeding about 10 m$^3$ s$^{-1}$ during the year 2010 (Fig. S7, S8).



## 4.2 Environmental noise pick-up of the geophone signal

Hydrophones (underwater microphones) have been used to monitor bedload transport both in riverine and in coastal environments (e.g. Thorne, 1990; Camenen et al. 2012; Basset et al., 2013). The objective of using such a system is to record self–generated noise produced by collisions of moving bedload particles against each other or against the bed. The application of this bedload surrogate measuring system can be impaired by other sources of noise, which may be caused by vessel traffic, marine seismic exploration, or underwater military operations. If the main interest is in the acoustic signal due to bedload transport, discounting for other sources of noise may be challenging and will also depend for example on the spatial distance and the dominant frequencies of the different acoustic sources (Hildebrand, 2009; Etter, 2012; Basset et al., 2013).

For the application of impact plates with acoustic sensors installed in a streambed there is very few experience with non-bedload transport related sources of noise that may compromise their usefulness. We have shown in section 3.3 that road traffic is a likely source of environmental noise producing a similarly strong signal at the SPG system as low–intensity bedload transport during periods with moderate discharges. This observation was made for our two study streams Fischbach and Ruetz, where in both cases the stream bed runs very close–by to roads, which are located only about half the stream–width away from the edge of the bed. We have checked the impulse counts recorded for SPG systems installed at mountain streams in Switzerland, particularly for low flow periods during winter time. There were generally very few impulses recorded at these sites, indicating that road traffic is not an important source of noise. At these sites roads with regular traffic are situated clearly farther away from the channel profile than at the two Austrian sites of this study: at the Navisence stream in Zinal (Ancey et al., 2015) about 45 m (or 3 times the stream width), at the Albula River in Tiefencastel (Rickenmann et al., accepted) about 30 m (or twice the stream width) to a road or about 15 m to a parking lot of a single building, and at the Avançon de Nant stream near Pont de Nant about 20 m (or 4 times the stream width). The SPG system at the Erlenbach stream in Switzerland (Rickenmann et al., 2012) is situated about 45 m away from a road; at this site we observed implausible impulse counts limited to very short time periods that were likely due to hikers or possibly game passing at the site.

At the Riedbach stream in Switzerland the geophone measuring site is situated at a water intake at an elevation of 1800 m a.s.l, with few direct sunshine and often freezing temperatures during winter time. The access road ends at the water intake and is not open to the public. For a seven year period from 2009 to 2015 geophone measurements showed no systematic relationship between $IMP$ and $Q$ for discharges $Q$ smaller than about 0.4 $m^3 \ s^{-1}$, but a considerable number of $IMP$ were recorded for $Q$ values as small as 0.05 $m^3 \ s^{-1}$ (Schneider et al., 2016). These discharge conditions are typical for the winter period, and it was hypothesized that ice transport or break-up may be mainly responsible for the impulse counts. Impulses may be typically as high as between 1 and 100 impulses for all seven plates and for 10 minute recording intervals. Calculating a mean impulse value per plate for $Q < 0.3 \ m^3 \ s^{-1}$ and including also zero values, this results in an average




duration of about 5 hours for one impulse to be registered at the Riedbach by one of the seven steel plates. This relatively
low occurrence frequency does not contradict the ice transport or break-up hypothesis.

## 5. Conclusions

The Fischbach and Ruetz gravel–bed streams are characterized by important runoff and bedload transport during the
snowmelt season. As a bedload surrogate measuring technique, the Swiss plate geophone (SPG) system has been installed in
2007 in both streams. During the six year period 2008 – 2013, 31(Fischbach) and 21 (Ruetz) direct bedload samples were
obtained in the two streams, and these measurements were analysed to obtain calibration relations for the SPG system at the
two sites.
As applied at many other SPG sites in the past, we first established calibration relations using total transported bedload
mass and the number of geophone impulses. A second way of analysing the geophone calibration measurements consisted in
using bedload transport rates and geophone impulse rates. For the Fischbach the second approach resulted in two power law
calibration relations, with different coefficients and exponents for small and large transport rates. The exponent was smaller
than one for small transport rates, and larger than one for larger transport rates. For the Ruetz data with essentially only
lower transport intensities, the power law relation derived from the Fischbach is also in reasonable agreement with the Ruetz
calibration measurements. The non-linear power law calibration relations are in qualitative agreement with the observed
coarsening of the bedload with increasing transport rates. According to findings from flume studies the signal response per
unit bedload mass increases for small grains up to grain size of approximately 40 mm, and decreases again for larger grains
with increasing particle size (Wyss et al., 2016b); this provides qualitative support for the existence of the two power law
relations. A similar behaviour could be observed only for the calibration measurements at the Urslau stream in Austria
(Kreisler et al., 2016). In contrast, calibration measurements from six other sites, including the Ruetz stream, do not show
evidence for the existence of similar two–range power law calibration relations.
Amplitude information from the geophone signal was recorded in minute intervals at the Fischbach and Ruetz by
summing impulse counts separately for different amplitude classes (so-called AH data). Since signal amplitude correlates
with grain size at several SPG sites (Wyss et al., 2016a, 2016b, 2016c), this information was used to estimate the grain size
distribution for the bedload samples from the Fischbach and Ruetz. It was found that the observed coarsening of the grain
size distribution with increasing bedload flux could be qualitatively reproduced from the geophone signal using the AH data.
For smaller discharges at the Fischbach and Ruetz, in particular during the winter time, it was found that many
implausible geophone impulse counts were recorded. Both SPG measuring sites are situated very close to local roads with
regular traffic. The roads are only about half the stream width away from the steel plates, and we therefore identified vehicle
traffic as a likely source for the implausible geophone impulses. This is indirectly supported by a comparison with other SPG
sites in Switzerland. At most of these sites only very few implausible geophone impulse counts were recorded in the past,



which is probably due to the fact that the local roads are farther away from the steel plates, generally at least once or twice
the stream width.
**6. Data availability**
The data cannot be made publicly available for the time being since it is used by the Hydropower Company TIWAG, the
owner and provider of the data, in an ongoing hydropower project authorisation procedure.
**7. Appendix A: Summary of the amplitude histogram method of Wyss et al. (2016a)**
Information about the grain–size distribution of the transported bedload over a Swiss geophone plate can be determined
using the number of impulses per amplitude class (called amplitude histogram method). Amplitude histograms (AH data)
can be interpreted as a statistical distribution of the signal's amplitude over a given time interval. Using the number of
bedload particles per unit mass, absolute bedload masses for each grain–size class were calculated for the Erlenbach stream
in Switzerland.
For j grain size classes an amplitude threshold value $A_{th}$ (upper class boundary value, in V) corresponds to a threshold
particle size $D$ in (mm) separating the grain size class (Wyss et al. 2016a). In this study an empirical relation given in Wyss
et al. (2016c) was used (see also Table 2):
$D = 85.5\,A_{th}^{0.41}$ (A1)
Wyss et al. (2016a) assumed that the number of impulses per amplitude class, $IMP_j$, are related to the number of particles in
the corresponding grain size class, $N_j$, with a mean weight, $G_{mj}$, by a coefficient $\alpha_j$ determined from the bedload samples, as
follows:
$IMP_j = \alpha_j\, N_j$ (A2)
For the calibration of the method for the Erlenbach 31 bedload samples were used. The analysis resulted in the following
empirical power law relation between $\alpha_j$ and the class mean grain size $D_{mj}$ in [mm] where the median value of $\alpha_j$ of all
bedload samples was used to determine the empirical relation (A3):
$\alpha_j = 0.0093\, D_{mj}^{1.09}$ (A3)
where the coefficient 0.0093 has the units [mm$^{-1.09}$]. Finally, to estimate the bedload mass per grain size class, the following
relation can be used:
$M_{est} = N_j\, G_{mj} = \dfrac{IMP_j\, G_{mj}}{\alpha_j}$ (A4)




The above procedure was used to estimate the bedload mass for each calibration sample from the Fischbach and the Ruetz,
as reported in section 4.1 in the Discussion. To determine the mean weight, $G_{mj}$ in [g] for each grain size class with $D_{mj}$ in
[mm], the following empirical relations were used, based on investigations reported in Wyss et al. (2016c):
$G_{mj} = 0.00165\ D_{mj}^{2.94}$            for the Fischbach                      (A5)
$G_{mj} = 0.00111\ D_{mj}^{3.03}$            for the Ruetz                          (A6)
Considering Eqs. (A5) or (A6) together with Eqs. (A2) and (A3) it follows that the number of grains per class is
approximately proportional to $IMP_j \bullet D_{mj}^2$. We used this proportionality in section 3.2 to estimate the GSD for the calibration
measurements from the Fischbach and Ruetz based on the recorded AH data. The main uncertainty in transferring the
method of Wyss et al. (2016a) determined for the Erlenbach to another site is the use of Eq. (A3) which may different at
other sites. We used the entire procedure reported here, including Eq. (A3) with the coefficient and exponent determined
from the Erlenbach measurements, in section 4.1 to explicitly estimate the total bedload mass for each calibration
measurement from the Fischbach and Ruetz based on the recorded AH data.
**8. Supplement link**
→ see also Supplementary Material
**9. Author contribution**
BF was the main responsible for the concept and installation of the SPG system at the Fischbach and Ruetz. He had
suggested to record the AH data as a memory efficient way to extract grain–size relevant information from the raw geophone
signal. DR was responsible for the analysis and wrote the paper. Support of colleagues for figure preparation is
acknowledged below.

**10. Acknowledgements**
We are grateful to the Tyrolean Hydropower Company (TIWAG) for having performed the geophone calibration
measurements in the Fischbach and Ruetz streams and for having provided these data and the continuous geophone
measurements to WSL for further analysis. The study was supported by SNF grants 200021_124634 and 200021_137681.
We thank Nicloas Steeb, Philipp von Arx, and Thomas Weninger for help with the preparation of some figures; TW also
performed grain size analyses of the streambed surface.



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




**Table 1.** Catchment and channel characteristics at the field sites and range of typical parameters for the conditions during the
geophone calibration measurements. The $q_b$ values refer to bedload with $D > 10$ mm.

|  | Fischbach | Ruetz |
|---|---|---|
| *Catchment parameters* | | |
| Drainage area (km$^2$) | 71 | 28 |
| Maximum elevation (m) | 3497 | 3474 |
| Site elevation (m) | 1540 | 1684 |
| Mean annual precipitation (mm) | 1670 | 1880 |
| % glacier | 16 | 20 |
| *Channel parameters (measuring site)* | | |
| Gradient over 60 m upstream of geophone site $S$ | 1.7 | 2.5 |
| Stream bed width (m) | 8.5 | 8.5 |
| Bed surface $D_{84}$ (m) | 0.26 | 0.28 |
| Bed surface $D_{50}$ (m) | 0.09 | 0.10 |
| *Parameter range for calibration periods* | | |
| Period of calibration measurements used in this | 2008–2013 | 2008–2013 |
| No. of calibration measurements used in this study | 31 | 21 |
| Max. unit discharge $q_{max}$ (m$^2$/s) | 1.97 | 0.97 |
| Min. unit discharge $q_{min}$ (m$^2$/s) | 0.56 | 0.41 |
| Max. mean flow velocity $V_{max}$ (m/s) | 2.79 | 1.88 |
| Min. mean flow velocity $V_{min}$ (m/s) | 1.51 | 1.02 |
| Max. unit bedload transport rate, $q_{b,max}$ (kg/sm) | 7.20 | 0.214 |
| Min. unit bedload transport rate, $q_{b,min}$ (kg/sm) | 0.0050 | 0.0025 |
| Bedload samples: max. $D_{max}$ (m) | 0.350 | 0.150 |
| Bedload samples: min. $D_{max}$ (m) | 0.030 | 0.050 |
| Bedload samples: max. weight (D > 10mm) [kg] | 431 | 128 |
| Bedload samples: mean weight (D > 10mm) [kg] | 70.0 | 20.6 |
| Sampling duration of calibration measurements [s] | 30 – 3600 | 600 – 3600 |
| Recording interval of geophone summary values | 900 | 900 |





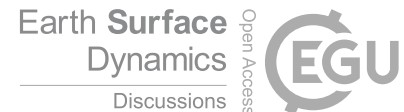


**Table 2**. Threshold values of the signal amplitude *A* used for the impulse count of the amplitude histograms at the Fischbach
and Ruetz. To estimate a corresponding particle size *D*, an empirical relation from Wyss et al. (2016c) was used.
$D_{mg}$ is the geometric mean size of each particle class.

| $A_{th}$ [V] | 0.056 | 0.079 | 0.112 | 0.158 | 0.224 | 0.316 | 0.447 | 0.631 | 0.891 | 1.259 | 1.778 | 2.512 | 3.548 | 5.012 | 7.079 | 10.0 | 12.0 |
|---|---|---|---|---|---|---|---|---|---|---|---|---|---|---|---|---|---|
| $D$ [mm] | 26.2 | 30.2 | 34.8 | 40.1 | 46.3 | 53.3 | 61.5 | 70.8 | 81.5 | 94.0 | 108 | 125 | 144 | 166 | 191 | 220 | 237 |
| $D_{mg}$ [mm] | | 28.1 | 32.4 | 37.4 | 43.1 | 49.7 | 57.2 | 66.0 | 76.0 | 87.5 | 101 | 116 | 134 | 154 | 178 | 205 | 228 |

555

556



557

**Table 3.** Coefficients, exponents and statistical properties for the calibration relations according to eq. 1, 2, 3, 4, 5. All calibration relations refer to bedload mass with $D > 10$ mm, or unit bedload transport rate $q_{b,p}$ for $D > 10$ mm. In the equations, the units are: $M$ in [kg], $q_{b,p}$ in [kg/0.5m/s] and $IMPT$ in [1/0.5m/s]. Here $r^2$ is the correlation coefficient between values calculated with the regression relation and the recorded bedload masses. Similarly, in all figures, $r^2$ is determined between the predicted y–value and the observed y–value (in the linear domain). The relative standard deviation $s_{e,r}$ is determined for the ratios ($M_{est}/M$) of estimated bedload mass $M_{est}$ calculated with the regression relation and the recorded impulses $IMP$, divided by the recorded bedload mass $M$. For the first three relations, the number of calibration measurements (n) are given in Table1, for the other two relations they are listed in this table.

567

|  | Fischbach | Ruetz | both streams |
|---|---|---|---|
| $M = k_{lin}\ IMP$ |  |  |  |
| $k_{lin}$ | 0.0508 | 0.0436 |  |
| $r^2$ | 0.964 | 0.597 |  |
| significance level: probability p | <0.0001 | <0.0001 |  |
| $s_{e,r}$ | 0.67 | 1.38 |  |
| $M = k_{pow}\ M^e$ |  |  |  |
| $k_{pow}$ | 0.134 | 1.40 |  |
| e | 0.88 | 0.42 |  |
| $r^2$ | 0.967 | 0.576 |  |
| significance level: probability p | <0.0001 | <0.0019 |  |
| $s_{e,r}$ | 0.78 | 0.92 |  |
| $M = k_{tot}\ IMP$ |  |  |  |
| $k_{tot}$ | 0.0558 | 0.0547 |  |
| $r^2$ | 0.964 | 0.597 |  |
| significance level: probability p | <0.0001 | <0.0001 |  |
| $s_{e,r}$ | 0.73 | 1.73 |  |
| $q_{b,p} = a_1\ IMPT^{b1}$    for IMPT < 0.48 [1/0.5m/s] |  |  |  |
| $a_1$ | 0.0237 | 0.0237 | 0.0237 |
| $b_1$ | 0.48 | 0.48 | 0.48 |
| $n$ | 15 | 15 | 30 |
| $r^2$ | 0.559 | 0.790 | 0.524 |
| significance level: probability p | <0.0001 | <0.054 | <0.0001 |
| $s_{e,r}$ | 0.77 | 1.13 | 0.98 |
| $q_{b,p} = a_2\ IMPT^{b2}$    for IMPT > 0.48 [1/0.5m/s] |  |  |  |
| $a_2$ | 0.0436 | 0.0436 | 0.0436 |
| $b_2$ | 1.29 | 1.29 | 1.29 |
| $n$ | 16 | 6 | 22 |
| $r^2$ | 0.964 | 0.517 | 0.966 |
| significance level: probability p | <0.0001 | <0.0001 | <0.0001 |
| $s_{e,r}$ | 0.37 | 1.71 | 1.11 |



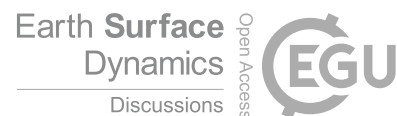

568

**Table 4.** Comparison of yearly bedload (*YBL*, in t) calculated with two different calibration relations, for the year 2010 and for different ranges of *Q* values. The *YBL* values represent transport over the entire stream width; as only every second steel plate is equipped with a geophone sensor, the loads inferred from the geophone impulses were multiplied by a factor of 2 in this table.


| Stream | Year | Q range | Yearly bedload YBL-A (t) Eqs. (4,5) | Yearly bedload YBL-B (t) Eq. (1) | YBL-A / YBL-B |
|--------|------|---------|---------|---------|---------|
| Fischbach | 2010 | all Q (including implausible IMP values) | 10,800 | 6,430 | 1.68 |
| | | Q > 3.5 m3 s-1 | 10,600 | 6,410 | 1.66 |
| Ruetz | 2010 | all Q (including implausible IMP values) | 1,360 | 621 | 2.19 |
| | | Q > 1.5 m3 s-1 | 1,110 | 600 | 1.85 |






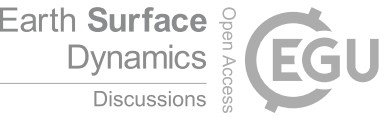

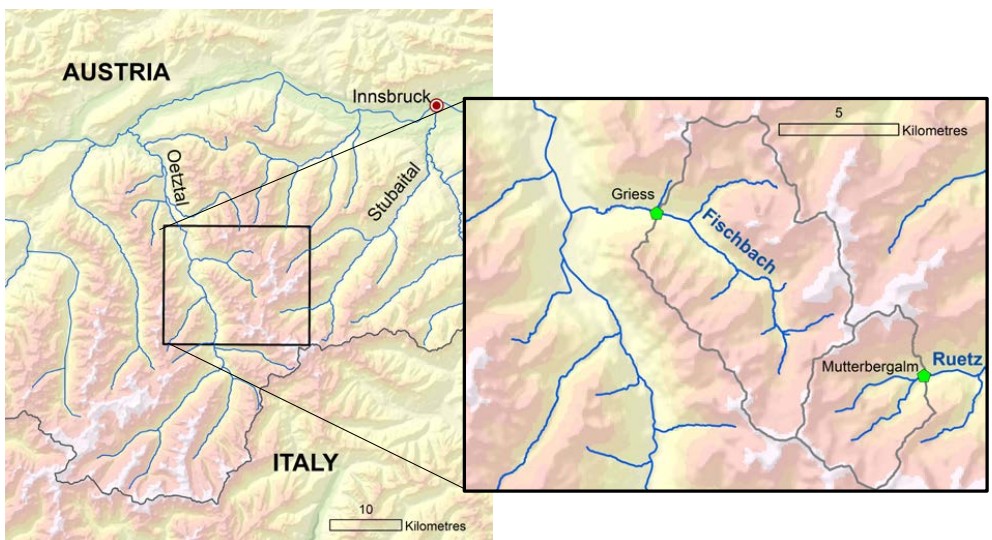

**Figure 1.** Location of the Fischbach and Ruetz mountain stream catchments in the Stubai Alps of Tyrol in western Austria. The measuring sites are indicated with a green pentagon, and the catchment boundaries are marked with a gray line. (Source of topographic map: Abteilung Geoinformation, Amt der Tiroler Landesregierung; https://www.tirol.gv.at/data)

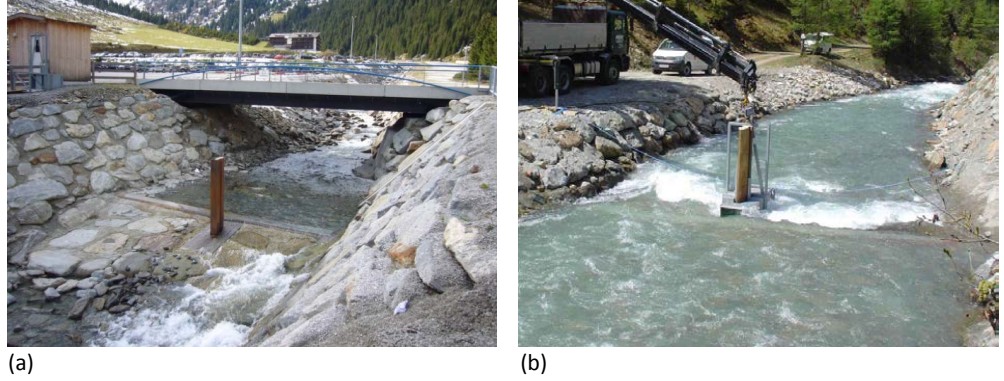

(a)               (b)

**Figure 2.** Monitoring sites equipped with a Swiss plate geophone system and a flow gauging station. (a) Ruetz, looking upstream onto the sill with the steel plates (28 October 2009), (b) Fischbach, looking downstream during a calibration measurement using the TIWAG basket sampler (27 May 2008). The steel-concrete pillar visible in both photos is used to guide the positioning of the basket sampler during the collection of bedload samples immediately downstream of the geophone plate.




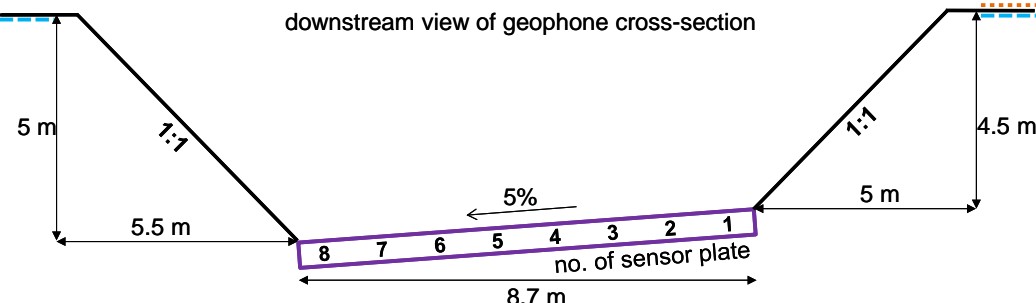

**Figure 3.** Schematic stream cross-section at the geophone measuring site in both Fischbach and Ruetz. The steel-concrete pillar is located downstream of the sensor plate no. 5. The sill with the steel plates is inclined towards the left bank to improve the resolution of the flow gauge measurements at low discharges. On the banks, the dotted horizontal line indicates the paved local road on river right side at the Fischbach, and the two dashed horizontal lines indicate the graveled parking lot on both river sides at the Ruetz.

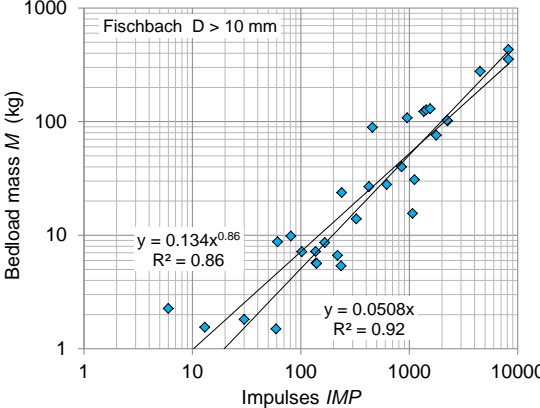
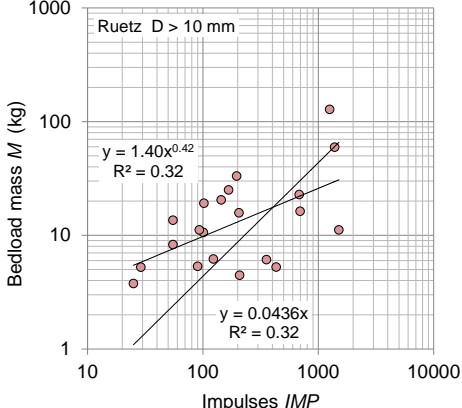

**Figure 4.** Fischbach: Geophone calibration relationships for grains with $D > 10$ mm between bedload mass $M$ and number of impulses $IMP$. The linear and power law regression equations are based on 31 calibration measurements for the years 2008-2013.

**Figure 5.** Ruetz: Geophone calibration relationships for grains with $D > 10$ mm between bedload mass $M$ and number of impulses $IMP$. The linear and power law regression equations are based on 21 calibration measurements for the years 2008-2013.





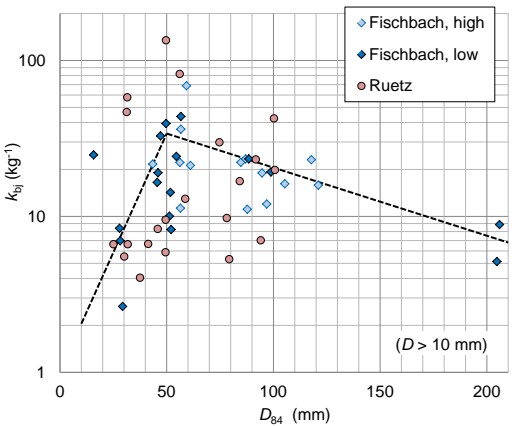

**Figure 6.** Linear calibration coefficient $k_{bj}$ versus characteristic grain size $D_{84}$, determined for particles with $D > 10$ mm. Fischbach: data points marked "high" and "low" refer to impulse rates higher and lower than 1 $(0.5^{-1}$ m$^{-1}$ s$^{-1})$, respectively. The dashed lines are meant to guide the eye.

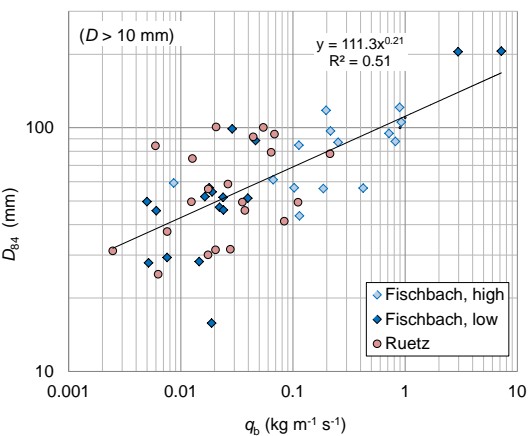

**Figure 7.** Characteristic grain size $D_{84}$ (determined for particles with $D > 10$ mm) versus bedload flux $q_b$, derived from the calibration bedload samples (for $D > 10$ mm). Fischbach: data points marked "high" and "low" refer to impulse rates higher and lower than 1 $(0.5^{-1}$ m$^{-1}$ s$^{-1})$, respectively. The regression line is based on both the Fischbach and Ruetz data.

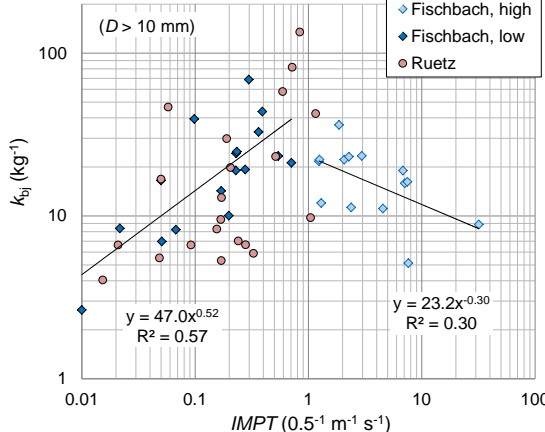

**Figure 8.** Linear calibration coefficient $k_{bj}$ (for $D > 10$ mm) versus impulse rate $IMPT$. Fischbach: data points marked "high" and "low" refer to impulse rates higher and lower than 1 $(0.5^{-1}$ m$^{-1}$ s$^{-1})$, respectively. The regression lines are based on the Fischbach data only.

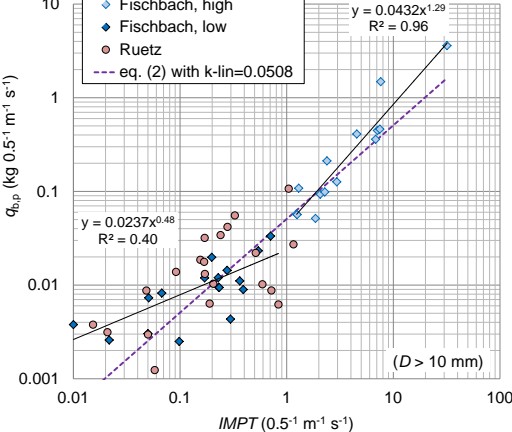

**Figure 9.** Unit bedload transport rate $q_{b,p}$ for particles $D > 10$ mm vs. impulse rate $IMPT$. Fischbach: data points marked "high" and "low" refer to $IMPT$ values higher and lower than 1 $(0.5^{-1}$ m$^{-1}$ s$^{-1})$, respectively. The regression lines are based on the Fischbach data only. The violet dashed line represents the linear calibration relation Eq. (2) determined for the Fischbach data based on a regression of $M$ vs. $IMP$.



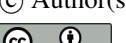

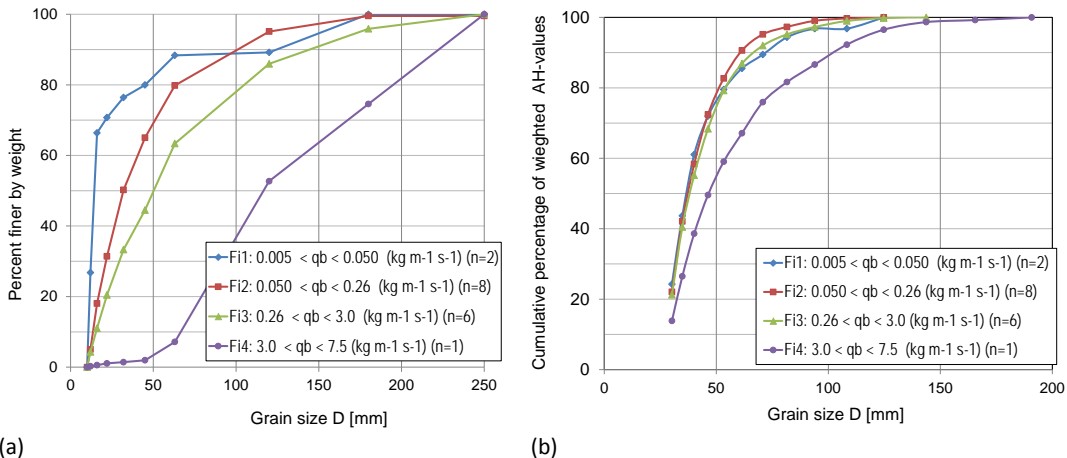

(a)        (b)

**Figure 10.** . Fischbach: (a) Grain size distributions derived from the calibration bedload samples (for $D > 10$ mm), averaged for four classes of bedload fluxes $q_b$ (using 17 samples from 2010-2012). (b) Relative distribution of grain sizes estimated from the geophone measurements based on the AH data, averaged for the same four classes of bedload fluxes (using the same 17 sample periods from 2010-2012).

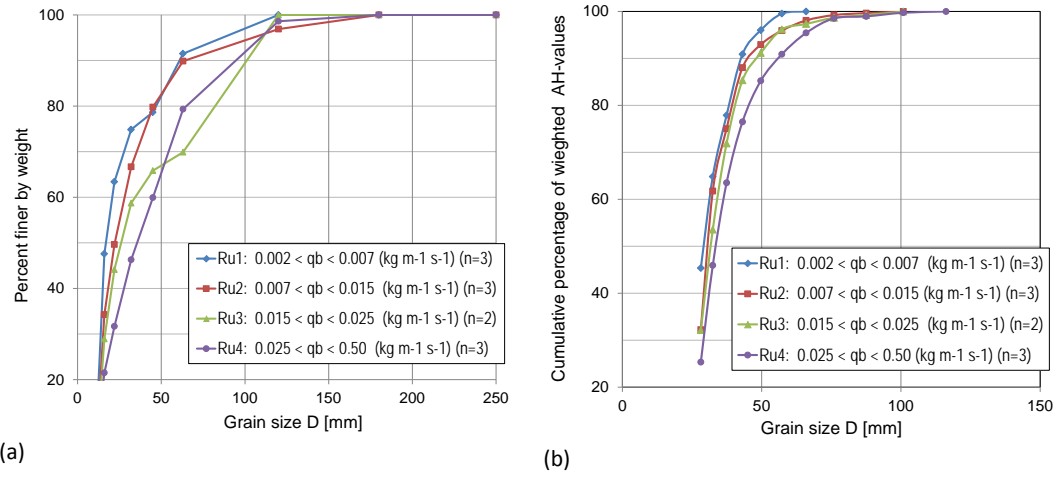

(a)        (b)

**Figure 11.** Ruetz: (a) Grain size distributions derived from the calibration bedload samples (for $D > 10$ mm), averaged for four classes of bedload fluxes $q_b$ (using 11 samples from 2010-2013). (b) Relative distribution of grain sizes estimated from the geophone measurements based on the AH data, averaged for the same four classes of bedload fluxes (using the same 11 sample periods from 2010-2013).

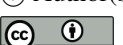



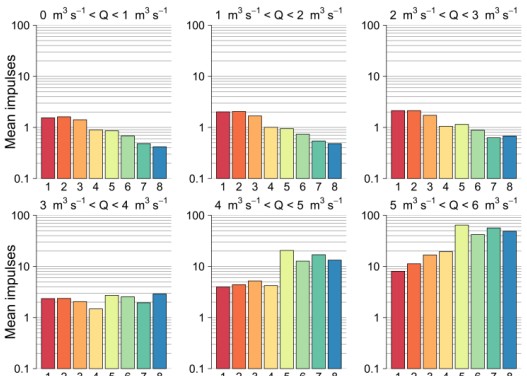

**Figure 12.** Fischbach: Arithmetic mean of geophone impulses per 15 min for each of the eight plates (ordinates), averaged over the period 2008-2013 and including zero values, for discharge $Q$ classes with width of 1 m$^3$ s$^{-1}$, for discharges up to 6 m$^3$ s$^{-1}$.

**Figure 13.** Ruetz: Arithmetic mean of geophone impulses per 15 min for each of the eight plates (ordinates), averaged over the period 2008-2013 and including zero values, for discharge $Q$ classes with width of 0.5 m$^3$ s$^{-1}$, for discharges up to 3 m$^3$ s$^{-1}$.

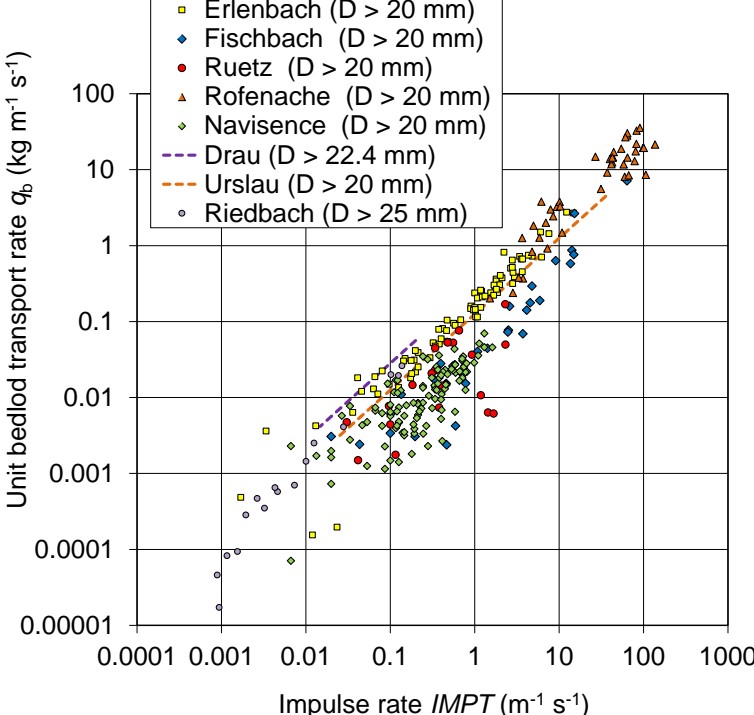

**Figure 14.** Comparison of geophone calibration data from eight different stream sites. Unit bedload transport rate $q_b$ for particles with $D$ larger than (mostly) 20 mm is plotted against impulse rate $IMPT$. Data sources for additional data are: Wyss et al. (2016c) for Navisence and Erlenbach (some data up to 2016 were added here); Habersack et al. (2016) for Drau; Kreisler et al. (2016) for Urslau (linear calibration relation is approximate; $q_b$ values given for $D > 10$ mm were reduced by factor of 0.68 to estimate $q_b$ values for $D > 20$ mm; reduction factor was estimated from 85 samples of Erlenbach moving basket data); Schneider et al. (2016) for Riedbach.





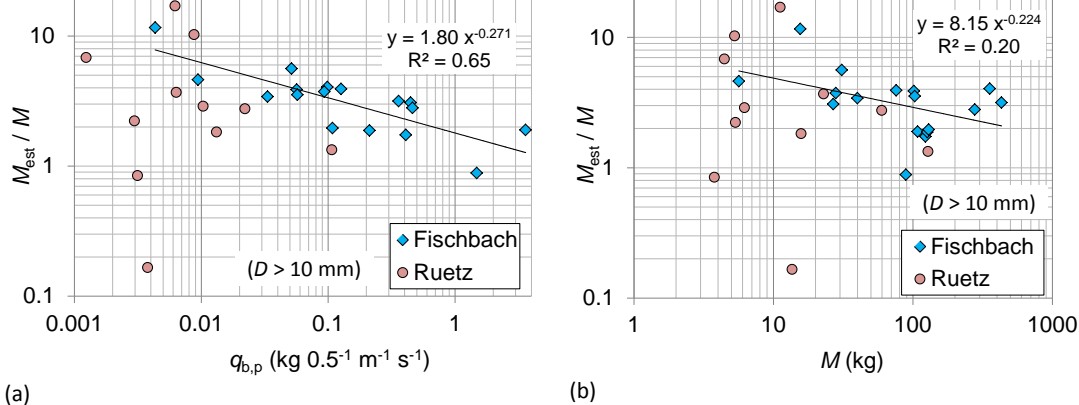

**Figure 15.** The estimated bedload mass per sample using the method in Wyss et al. (2016a) developed for the Erlenbach, $M_{est}$, is compared with the measured bedload mass, $M$, through the ratio $M_{est}/M$. (a) Ratio $M_{est}/M$ shown vs. unit bedload transport rate $q_{b,p}$ for particles with $D > 10$ mm, (b) Ratio $M_{est}/M$ shown vs. measured bedload Mass $M$ for particles with $D > 10$ mm. In both diagrams the regression line is based on the Fischbach data only.