# Peer review of "Bedload transport measurements with impact plate geophones in two Austrian mountain streams (Fischbach and Ruetz): system calibration, grain size estimation, and environmental signal pick-up"

_Earth Surface Dynamics, 2017_

## Referee Comment (RC2) · Anonymous Referee #2 · 28 May 2017

The authors present reports on calibration measurements of the Swiss plate geophone (SPG) system in two mountain streams, the Fischbach and Ruetz gravel-bed in Austria. A total of 52 measurements were recorded. These streams are characterized by important runoff and bedload transport during the snowmelt season. The paper covers: different ways of analyzing the geophone calibration measurements, how the observed coarsening of the grain size distribution with increasing bedload flux can be qualitatively reproduced from the geophone signal, and the geophone impulse noises. Lots

of statistical analyses are presented to correlate between the bed load mass and geophone impulse. Such a detailed and technical analysis can be important in hydraulic applications, including, particle-laden stream flows and power plants. This is a largely technical paper; well written, well organized and discussed. The audience of Earth Surf. Dynam. may benefit from its publication. However, it is not clear if the method is also applicable for more dense flows as a mixture of viscous fluid and particles including the debris flows, debris floods and other types of particle transports. It would be relevant to discuss these and similar aspects.

Title: Do you need to mention about SPG and stream names in the title? Could be made more elegant?

Detailed comments/suggestions:

- L72: what is the slope in flow direction?
- L118: IMP -> IMP (Impulse)
- L161: a slightly poorer performance: elaborate.

L175-176: Notation: IMPT sounds a bit strange. Is T here for 'Transport rate'? Then, change to IMPR? Also why not to avoid  $0.5^{-1}$ ?

L231-232: traffic noise appears to be a likely source of the geophone impulses: Couldn't that be checked by running some vehicles over there?

L265-266: it is known that the signal response depends on factors such as grain size, fluid or particle velocity, particle shape and mode of transport: So, the system could also be potentially used for debris flows/floods as in Mergili et al. (2017, GMD), von Boetticher et al. (2016, GMD), Pudasaini (2012, JGR), etc. May be discuss on this.

Table 1: Why do you use different 'Sampling duration of calibration measurements'?

Table 3: (a1, b1) and (a2, b2) are the same for both sites. What does it mean?

**ESurfD**
Fig. 6: The dashed lines are meant to guide the eye.: Not clear how?

Fig. 14: Lines in Fig.: are these regression lines?

**ESurfD**

---

## Author Comment (AC2) · 15 Jun 2017

**"Bedload transport measurements with Swiss impact plate geophones in two Austrian mountain streams (Fischbach and Ruetz): system calibration, grain size estimation, and environmental signal pick-up"**

by Dieter Rickenmann and Bruno Fritschi

Swiss Federal Research Institute WSL, Birmensdorf, 8903, Switzerland

A revised version of the manuscript is attached, indicating the modified parts in red.

Reply to Comments of Reviewers are given below.

**Anonymous Referee #1**

We thank the reviewer for his/her constructive comments. Below are the responses to these comments and an indication on how we made related changes in the manuscript.

**General comments**

**This paper reports measurement results of bedload transport with Swiss plate geophones in two mountain streams in Austria. The authors focused on calibration, grain size estimation and noises included in the recorded data. Their calibration was carried out by direct sampling. The results help researchers who are working on similar topics at different streams. The description of the environment noise (3.3) is reasonable and interesting.**

**Specific comments**

**Please show the basket bedload sampler by a figure or a photo**

A drawing of the sampler was already included in the Supplement information to the paper. We added two photos in the Supplement information for further illustration of the sampler.

**Please show the grain size distribution of the bed surface measured by a line-by–number analysis.**

We added a figure in the manuscript showing the grain size distribution of the bed surface upstream of the geophone sites for both streams.

**Is there any hysteresis about grain size and bedload flux relationship?**

Based on the analysis of the grain size distributions (GSD) of all bedload samples we found on average a coarsening of the GSD with increasing bedload transport intensity (Figures 7, 10). However, as mentioned in the manuscript, GSDs from individual calibration measurements (bedload samples) are quite variable within given classes of bedload transport rates. The same is true if GSDs of the bedload samples are analyzed in terms of changing discharge. The bedload samples were taken too randomly in time and too infrequently over the six years study period as to allow to examine whether there is any hysteresis trend for daily discharge cycles or over the entire summer season. In a follow-up study, possible hysteresis trends were investigated based on the continuous geophone data which were converted into bedload fluxes using equations (4) and (5), and the related findings will be discussed in an upcoming paper.

**It is difficult to understand; lines 257-262.**

Turowski et al. (2011) determined discharge values at the start ($Q_s$) and at the end ($Q_e$) of a transport period for the Fischbach and the Ruetz streams. The $Q_s$ and the $Q_e$ values that are smaller than the $Q_c$ values identified in our manuscript for the two streams, respectively, may contain implausible impulse counts. In other words, some of the $Q_s$ and the $Q_e$ values identified in the study of Turowski et al. (2011) may not represent start and end of bedload transport. The text in the manuscript was modified to clarify the meaning of the paragraph.

**Technical corrections. What is 'nival' at line 69?**

This statement was imprecise, it was changed to: "Thus, the streams have a nival and glacial runoff regime, with typical daily discharge variations and regular bedload transport during snow and glacier melt in spring and summer." The term "nival" refers to snow-related processes.

**Anonymous Referee #2**

We thank the reviewer for his/her constructive comments. Below are the responses to these comments and an indication on how we made related changes in the manuscript.

**The authors report on calibration measurements of the Swiss plate geophone (SPG) system in two mountain streams, the Fischbach and Ruetz gravel-bed in Austria. A total of 52 measurements were recorded. These streams are characterized by important runoff and bedload transport during the snowmelt season. The paper covers: different ways of analyzing the geophone calibration measurements, how the observed coarsening of the grain size distribution with increasing bedload flux can be qualitatively reproduced from the geophone signal, and the geophone impulse noises. Lots of statistical analyses are presented to correlate between the bed load mass and geophone impulse. Such a detailed and technical analysis can be important in hydraulic applications, including, particle-laden stream flows and power plants. This is a largely technical paper; well written, well organized and discussed. The audience of Earth Surf. Dynam. may benefit from its publication. However, it is not clear if the method is also applicable for more dense flows as a mixture of viscous fluid and particles including the debris flows, debris floods and other types of particle transports. It would be relevant to discuss these and similar aspects.**

**Title: Do you need to mention about SPG and stream names in the title? Could be made more elegant?**

We have modified the title by changing the term "Swiss impact plate geophones" to "impact plate geophones".

**Detailed comments/suggestions:**

**L72: what is the slope in flow direction?**

The steel plates are horizontal in flow direction (no longitudinal slope). The riprap on the downstream side of the sill is inclined at about 15% over a length of about 2 m. This information was added in the manuscript.

**L118: IMP –> IMP (Impulse)**

Thank you, this definition was added in the manuscript.

**L161: a slightly poorer performance: elaborate.**

We noticed that the relative clause "also with a slightly poorer performance (Table 3)" was not very clear. We changed the formulation in the manuscript to: "while the relative standard deviation $s_{e,r}$ is very similar for the Fischbach in both cases, it is about 25% larger for the Ruetz when using the $k_{tot}$ coefficient as compared to using the $k_{lin}$ coefficient (Table 3)."

**L175-176: Notation: IMPT sounds a bit strange. Is T here for 'Transport rate'? Then, change to IMPR? Also why not to avoid 0.5ˆ{-1}?**

In the abbreviation IMPT the letter T stands for "Time" or "per Time". It is a matter of definition, we prefer to leave it as it is. We admit that unit $0.5^{-1}$ looks somewhat awkward. However, we prefer to keep it this way, for the reason stated in the manuscript (risk of erroneous applications).

**L231-232: traffic noise appears to be a likely source of the geophone impulses: Couldn't that be checked by running some vehicles over there?**

Thank you, this is a very good suggestion. Indeed, such an "experiment" could and should be done during a period of low flow (winter) when there is no bedload transport activity.

**L265-266: it is known that the signal response depends on factors such as grain size, fluid or particle velocity, particle shape and mode of transport: So, the system could also be potentially used for debris flows/floods as in Mergili et al. (2017, GMD), von Boetticher et al. (2016, GMD), Pudasaini (2012, JGR), etc. May be discuss on this.**

We agree that similar acoustic methods may be useful to monitor debris flows. However, according to our experience, there may be some limitations when extrapolating calibration relations for the SPG system from the typical range of conditions investigated so far to extreme flow conditions and very high bedload transport rates. This issue was not discussed in earlier publications of the Swiss plate geophone system, and therefore we added a paragraph on this topic in the first part of the Discussion section, and we included a new diagram in the Supplement information.

**Table 1: Why do you use different 'Sampling duration of calibration measurements'?**

The sampling duration was essentially selected according to bedload transport intensity. For very high bedload transport rates, the sampler may be quickly filled; ideally, sampling should be stopped before the basket is full (e.g. scouring of previously caught particles, uncertainty about exact filling time). For very small bedload transport rates, the total sampled mass may be relatively small for a fixed sampling duration; if only few particles travel over the steel plate, the variability of the signal response is larger, due to random factors influencing the signal response (e.g. different transport modes and impact locations) that only tend to average out for larger numbers of particles that moved over the plate (see also manuscript L265-L274). Therefore we used generally longer sampling durations for lower transport rates.

**Table 3: (a1, b1) and (a2, b2) are the same for both sites. What does it mean?**

The reason for this is evident from Figure 9. For the lower bedload transport or impulse rates (range of validity of Eq. 4), the calibration measurements from the two streams show a very similar trend. For the higher bedload transport or impulse rates (range of validity of Eq. 5), there are only calibration measurements from the Fischbach. The basic assumption here is that Equations (4) and (5) are valid for both streams.

**Fig. 6: The dashed lines are meant to guide the eye: Not clear how?**

Essentially, the dashed lines were drawn to illustrate a trend for $k_{bj}$ values to increase with increasing $D_{84}$ values up to an "optimum" grain size, and to decrease again for further increasing grain sizes, because this pattern is in agreement with a similar pattern observed in systematic flume experiments (Wyss et al., 2016b; Rickenmann et al., 2014). This information is contained in L163-L168 in the manuscript.

**Fig. 14: Lines in Fig.: are these regression lines?**

Yes, these are power law regression lines based on the Fischbach data.

[revised manuscript text omitted]

**Figure S9.** Data with piezoelectric bedload impact sensors (PBIS) measurements made at a water intake of the Pitzbach mountain stream in Austria during two summer periods (Rickenmann and McArdell, 2008). Impulses were counted in a similar way as for the Swiss plate geophone system. Here, impulses and bedload volumes were aggregated over daily periods. At the Tyrolean weir a total of 12 steel plates with sensors were installed, with a natural gravel-bed surface upstream of the sill of 6 m width. At the flushing canal, only 3 steel plates with sensors were installed, at the end of a 1.5 m wide concrete channel. Flushing of sediment from the settling basin occurred over relatively short time periods and thus produced high velocity flows and much higher bedload concentrations in the flow than at the (natural) approach flow to the Tyrolean weir. While a reasonably well defined calibration relation could be obtained for the measurements at the Tyrolean weir (Rickenmann and McArdell, 2008), a very large scatter can be observed for the calibration data of the flushing canal, for bedload volumes smaller than 100 m³ (see above figure). This observation indicates that there are limitations for the SPG system for extreme flow conditions. The PBIS measurements at the Pitzbach mountain stream were made by the Tyrolean Hydropower Company (TIWAG).

---

## Author Response (AR2)

**"Bedload transport measurements with Swiss impact plate geophones in two Austrian mountain streams (Fischbach and Ruetz): system calibration, grain size estimation, and environmental signal pick-up"**

by Dieter Rickenmann and Bruno Fritschi

Swiss Federal Research Institute WSL, Birmensdorf, 8903, Switzerland

A revised version of the manuscript is attached, indicating the modified parts in red.

Reply to Comments of Reviewers are given below.

We thank the AE for his further comments helping to improve the manuscript. Below are the responses to these comments.

**Associate Editor Decision**: Publish subject to technical corrections (07 Sep 2017) by Michael Dietze

**The replies to the referee comments appear all very appropriate and justified. From reading the revised manuscript and referee communications I suggest to address the comment by referee 1 about the potential hysteresis pattern and comments by referee 2 regarding table 1 (different sampling duration of calibration measurements), table 3 (a1, b1 and a2, b2 are the same for both sites) and new figure 15 (dashed lines are regression lines) not only in the replies to the comments but also in the manuscript.**

- A new paragraph has been included to add the information on the sampling duration (new L 115-122).
- Some lines have been added to further explain the choice of coefficients and exponents in Table 3 (new L 192-196).
- A new paragraph has been included to shortly address the issue of potential hysteresis patterns (new L 350-356).
- In the caption to the new Figure 15, the words "regression line from" have been added in front of "Habersack et al. (2017)" and in front of "Kreisler et al. (2017)"

**Likewise, make a link to the photos/sketches of the basket sampler from the manuscript (e.g., beginning of chapter 2.2).**

The new photos were added to Figure S1. Therefore, the already existing reference to this Figure S1 in the manuscript (new L 100) should be sufficient.

**Please also update the title for the supplementary materials.**

The title has been updated.

**Regarding the point about L265-266 by referee 2 ("...So, the system could also be potentially used for debris flows/floods..."), it may be useful -- in addition to the current discussion of own work -- to also give a short introduction of the findings by other studies, e.g., some of the ones mentioned by the referee.**

We think that the new paragraph added in the first revision (new L 306-324) is sufficient to address the issue raised by referee 2. It is evident from this discussion that the impact plate geophone system cannot be easily applied to high concentration flows such as debris floods or debris flows. We do not think that further references are helpful to clarify this point (we do not know of related applications other than the ones already cited).

[revised manuscript text omitted]